# Investigation of Serum Endocan Levels in SARS-CoV-2 Patients

**DOI:** 10.3390/ijms25053042

**Published:** 2024-03-06

**Authors:** Laura Constantin, Anca Ungurianu, Anca Streinu-Cercel, Oana Săndulescu, Victoria Aramă, Denisa Margină, Isabela Țârcomnicu

**Affiliations:** 1National Institute of Infectious Diseases “Prof. Dr. Matei Bals”, 021105 Bucharest, Romania; constantin.sterian@drd.umfcd.ro (L.C.); anca.streinucercel@umfcd.ro (A.S.-C.); dr.arama@mateibals.ro (V.A.); isa.tarcomnicu@gmail.com (I.Ț.); 2Department of Biochemistry, Faculty of Pharmacy, “Carol Davila” University of Medicine and Pharmacy, 020956 Bucharest, Romania; anca.ungurianu@umfcd.ro

**Keywords:** endocan, COVID-19, endothelial dysfunction

## Abstract

Endocan is an endothelial-cell-specific proteoglycan (ESM-1) and has emerged as an endothelial dysfunction and inflammatory marker in recent years. Endocan can be used as a marker of inflammatory endothelial dysfunction in endothelium-dependent disease: cardiovascular disease, sepsis, lung and kidney disease and malignancies. Recent data suggest that endothelial dysfunction is a key mechanism in COVID-19 pathogenesis. Endotheliitis and thrombo-inflammation are associated with severe forms of SARS-CoV-2 infection, and endocan is currently under investigation as a potential diagnostic and prognostic marker. The aim of this study was to determine serum endocan levels in patients with COVID-19 to evaluate the correlation between endocan levels and clinical disease diagnosis and prognosis. This study enrolled 56 patients, divided into three groups depending on disease severity: mild (15), moderate (25) and severe (16). The biochemical, demographic, clinical and imagistic data were collected and evaluated in correlation with the endocan levels. Serum endocan levels were significantly higher in the COVID-19 patients compared to the control group; also, endocan concentration correlated with vaccination status. The results revealed significantly elevated serum endocan levels in COVID-19 patients compared to the control group, with a correlation observed between endocan concentration and vaccination status. These findings suggest that endocan may serve as a novel biomarker for detecting inflammation and endothelial dysfunction risk in COVID-19 patients. There was no significant relationship between serum endocan levels and disease severity or the presence of cardiovascular diseases. Endocan can be considered a novel biomarker for the detection of inflammation and endothelial dysfunction risk in COVID-19 patients.

## 1. Introduction

Coronavirus disease 2019 (COVID-19) is a clinical syndrome caused by infection with a strain of coronavirus (SARS-CoV-2), called severe acute respiratory syndrome (SARS)-associated coronavirus, and is linked, for an important percentage of hospitalized patients, to long-term consequences and mortality [1]. Viral infection induces a dysregulated hyperactivation of the immune system, a prothrombotic state manifested as micro-thrombosis, disseminated inflammation and vascular endothelium injuries, mainly in the pulmonary area [2,3]. Although COVID-19 was initially considered a viral pneumonia leading to acute respiratory failure, the clinical, laboratory and postmortem findings suggest that altered endothelial function is a contributing factor of its systemic pathophysiology [4]. SARS-CoV-2 infection causes endothelial dysfunction by direct viral effect, through cytokine release, oxidative stress, coagulation disturbance and inflammation with leucocyte adhesion and dysregulated immune cell response [4].

Profound inflammatory response and endothelial activation/damage cause coagulopathy of COVID-19, a combination of low-grade disseminated intravascular coagulation and pulmonary thrombotic microangiopathy with high levels of D dimers and decreased platelet count [5].

Cardio-metabolic pathology (hypertension, diabetes, obesity, dyslipidemia), cardiovascular disease (heart failure, cardiac arrhythmias and conduction disorders, valvular disease), chronic respiratory disease and chronic kidney disease were associated with a higher risk of COVID-19 hospitalization and mortality [5,6]. Patients with cardiovascular comorbidities manifest a chronic vascular injury predisposing to SARS-CoV-2 infection with the involvement of the common inflammatory pathways. Viral infection induces endothelial dysfunction through activating coagulation pathways, followed by a coagulation imbalance. Elevated D dimers, increased fibrinogen and enhanced platelet activation indicate a dysfunctional endothelial response to viral infection [7]. Serum levels of interleukin 6 (IL6) are elevated in inflammatory conditions and are associated with COVID-19 severity and the presence of respiratory complications [8,9]. The activation of IL6 receptors induces the upregulation of adhesion molecules with enhanced leucocyte adherence and extravasation in vascular walls [10]. An increased C reactive protein (CRP) plasma concentration correlates with the severity and poor prognosis of COVID-19 [11]. Elevated levels of pro-inflammatory cytokines increase endothelial permeability and the expression of adhesive molecules with endothelial activation and dysfunction [4,11]. Immune dysregulation (cytokine storm), intravascular thrombosis, activation of reactive oxygen species and leukocyte adhesion result in endotheliitis/endotheliopathy [11].

Endocan (endothelial-cell-specific molecule 1, or ESM-1) is a molecule with a proteoglycan structure, secreted by endothelial cells based on the information from the endothelial-cell-specific molecule-1 (ESM-1) gene [12,13]; its secretion is stimulated as an effect of proinflammatory molecules (tumor necrosis factor (TNF)-α, interleukin (IL)-1, E-selectin) and also some other cellular mediators of lesional signals (bacterial lipopolysaccharide (LPS), angiogenic factors, vascular endothelial growth factor (VEGF) or fibroblast growth factor ((FGF)-2). Endocan’s main function is stimulating the transport of inflammatory molecules at endothelial cell levels, as well as regulating cell adhesion, migration, proliferation and neo-vascularization, thus playing a regulatory role in endothelial activation and dysfunction, being considered an emerging marker of cardiovascular disease [14].

In healthy subjects, low levels of endocan are present, but in pathological endothelium-dependent conditions, the plasma levels increase and can serve as a potential endothelial function marker in endothelial dysfunction pathologies such as cardiovascular diseases: atherosclerosis, hypertension or angiogenic processes such as cancers with vascular endothelial involvement. Several studies have investigated the relationship between endocan and pulmonary diseases. The peripheral blood endocan level was found to be increased in pneumonia, acute respiratory distress syndrome and venous thromboembolism [14].

Endocan can be used as a marker of inflammatory endothelial dysfunction in endothelial-dependent disease: cardiovascular disease, sepsis, lung and kidney disease and malignancies, etc. Kosir et al. suggest the utility of endocan levels as a biomarker for risk stratification in heart failure chronic patients [13]. Recent studies observed that elevated levels of circulating endocan correlated with inflammatory conditions and thrombotic events in COVID-19 patients [15]. Accumulating evidence suggests that COVID-19 is a microvascular and endothelial disease caused directly by viral infection and indirectly via the cytokine storm. Endothelial dysfunction includes different mechanisms such as oxidative stress, hyperpermeability, endothelial injury, inflammation with leucocyte adhesion and hypercoagulability followed by thrombosis [4]. Thrombo-inflammation and endotheliitis are associated with severe forms of SARS-CoV-2 infection, and endocan is a potential diagnostic and prognostic marker. 

Prior research has identified endocan as an indicator of endothelial dysfunction and inflammation in various endothelial-dependent diseases, but its relevance in COVID-19 should be studied. In this context, endocan should be investigated as a potential marker of endothelial damage, and its role in inflammatory and thrombotic events should be elucidated. The aim of our study was to evaluate the relationship between endocan levels in COVID-19-hospitalized patients and disease diagnosis and prognosis, as well as with the biochemical profile of patients and their vaccination status, in order to establish the relevance of endocan assessment as a prognostic marker for SARS-CoV-2 patients. This analysis contributes to the evidence regarding endocan’s role in inflammation and endothelial dysfunction in COVID-19.

## 2. Results

### 2.1. Clinical Characteristics of COVID-19 Patients

The study included 56 COVID-19 patients and 23 healthy controls without comorbidities. The demographic and clinical characteristics of the patients are summarized in Table 1. In the COVID-19 group, the mean age was 71.48 ± 14.75 years. The majority of patients (82.14%) had comorbidities, mostly cardiovascular disease (hypertension, heart failure, heart rhythm disturbances) with a rate of 64.28%. The proportion of subjects with SARS-CoV-2 also affected by diabetes was similar to those with hyperlipidemia (17.85%). Treatment with oxygen or dexamethasone was used for 38 patients. Thromboprophylaxis was used in moderate and severe cases with LMWH or new oral anticoagulant (apixaban). Pulmonary changes were observed on computed tomography (CT) images as interstitial pneumonia (44.64%) and interstitial pneumonia with pleural effusion/mediastinitis (32.14%). 

The majority (96.42%) of the patients were discharged after recovery; 3.57% died. 

### 2.2. Laboratory Findings

Table 2 displays the laboratory findings on admission for COVID-19 patients. Hyperglycemia was found in 50% of the patients. Lactate dehydrogenase (LDH) was elevated in 20 patients (35.71%), but alkaline phosphatase was normal in 89.29%. Biomarkers of inflammation were increased: CPR for 92.85% and fibrinogen for 44.64% of patients. Decreased levels of platelets were recorded for 46.42%, and D dimers were elevated in 69.64% of patients.

### 2.3. Stratification of COVID Patients

Patients were classified depending on disease severity from mild (stage 0) to severe (stage 3), based on criteria adapted from the NIH COVID-19 Treatment Guidelines. The numbers of patients were as follows: 15 patients in the mild group, 25 patients in the moderate group and 16 patients in the severe group. The demographic, clinical characteristics and the laboratory data of patients are summarized in Table 3, depending on the severity of the disease. There was a significant difference between the mean age of patients with mild and severe forms (63.5 vs. 83, *p <* 0.001). In the severe group, all patients suffered from cardiovascular disease and other pathologies such as obesity or diabetes. Vaccinating status differed between the groups; the proportion of unvaccinated subjects was higher in the mild and severe groups compared to the moderate one. Oxygen therapy was used in 81.25% of severe cases and only in one patient (4%) in moderate cases (Table 3).

Markers suggesting inflammation, such as CRP, were significantly higher in the severe group compared to the moderate group (*p* = 0.008) (Figure 1). Serum concentration of CRP correlated significantly with the severity of disease, presence of cardiovascular disease (*p* = 0.006), need of oxygen therapy (*p* < 0.0001) and computer-scan severity (*p* = 0.033). For patients with cardiovascular disease, CRP levels positively correlated with AST (r = 0.44, *p* = 0.006), ALT (r = 0.334, p = 0.043), LDH (r = 0.428; *p* = 0.011), IL6 (r = 0.704, *p* = 0.003) and fibrinogen (r = 0.602, *p* < 0.0001).

LDH, a marker suggesting cell injury, had significantly higher levels in the severe COVID-19 group compared to the mild one (*p* = 0.035) (Figure 2). The increase in the severity of CT thorax results (pneumonia with mediastinitis/pleural effusion) was associated with CRP values (*p* = 0.022). In severe cases, LDH correlated with IL6 (r = 0.714, *p* = 0.047), WBC (r = 0.598, *p* = 0.024) and fibrinogen (r = 0.620, *p* = 0.018). In subjects with cardiovascular comorbidities, serum LDH was significantly higher (*p* = 0.003). Patients with hypoxemia had LDH levels significantly higher compared to those without an oxygen-therapy need (*p* < 0.005).

Severe COVID-19 cases were associated with elevated levels of fibrinogen (*p* = 0.021) (Figure 3) and IL 6 (*p* = 0.012) more than moderate cases. In COVID-19 patients with cardiovascular disease, fibrinogen and IL6 levels positively correlated (r = 0.851, *p* = 0.000). Coagulation markers and D dimers were significantly higher in the subjects with severe forms of COVID19 compared to mild ones (*p* = 0.046) (Figure 4). D dimers correlated with severity in CT results (*p* = 0.033), and elevated levels 3-fold higher than the normal upper limit were associated with interstitial pneumonia with mediastinitis /pleural effusion. The serum levels of D dimers correlated with disease severity (*p* = 0.046) and oxygen-therapy need (p = 0.033). In patients with oxygen therapy, D dimers positively correlated with IL6 (r = 0.786; *p* = 0.036). In the group of unvaccinated patients, D dimers correlated with LDH (r = 0.393; *p* = 0.032). In severe cases, IL6 levels correlated with LDH (r = 0.714, *p* = 0.047), and in patients with oxygen therapy, they correlated with D-dimer levels (r = 0.786, *p* = 0.036) and fibrinogen (r = 0.851, *p* = 0.000).

Endocan serum levels in the severe group were comparable to those measured in the patients with moderate forms and did not exhibit a statistically significant difference (*p* = 0.117).

Cardiovascular diseases were the most common comorbidities, being observed in all patients with severe forms. The three patient groups were divided into two subgroups based on the presence/absence of cardiovascular diseases. Alkaline phosphatase, lactate dehydrogenase, C reactive protein and fibrinogen were significantly higher in the COVID-19 patients with cardiovascular diseases (Table 4). C reactive protein values positively correlated with LDH (r = 0.428; *p* = 0.011); IL6 (r = 0.704, *p* = 0.003) and fibrinogen (r = 0.602, *p* = 0.000).

### 2.4. Evaluation of Serum Endocan Levels

Endocan levels ranged between 10.87 pg/mL and 154.95 pg/mL in the subjects included in the study group, and its levels were significantly different (*p* < 0.001) in patients compared to controls. The median endocan level was 77.21 ± 31.46 pg/mL for the study group and 33.09 ± 11.76 pg/mL for the control one. For patients discharged after recovery, the median endocan value was 76.36 pg/ mL, but in those who died, it was 85.02 pg/mL.

Serum levels of endocan were not significantly different between patients with and without comorbidities (cardiovascular disease, obesity, diabetes), even if a certain tendency for increase was observed in patients with cardiovascular diseases. Significant differences were observed between vaccinated and unvaccinated subjects (Table 5). Unvaccinated patients had higher serum levels of endocan compared to vaccinated ones. In the unvaccinated patients, endocan levels negatively correlated with IL1 (r = 0.0636, *p* = 0.048). Further, patients with dexamethasone treatment (initiated at admission) had higher serum levels of endocan compared to those without corticoids, but the difference was not statistically significant. In patients from the mild group, endocan positively correlated with CKMB (r = 0.510, *p* = 0.04), alkaline phosphatase (r = 0.515, *p* = 0.03) and fibrinogen (r = 0.524, *p* = 0.03), and in patients without changes in thorax CT, it positively correlated with alkaline phosphatase (r = 0.537, *p* = 0.03). 

In this study, we did not record dosages for inflammatory secretory phospholipase, sPLA2, but the analysis of this parameter could be the subject of a future study. Lipase values were determined, but there was no correlation with endocan levels. According to Farooqui et al. (2023), sPLA2 levels correlate with severity in COVID-19, and the activity of type sPLA2 IIA is linked to cytokine storm and coagulopathy. There are no studies about the sPLA2–endocan relation, but it is a very interesting subject for future research, particularly as both molecules activate in endothelial cells [16].

## 3. Discussion

Numerous studies addressing COVID-19 pathophysiology have been published, but the disease mechanism is still not completely elucidated, especially since reports regarding COVID-19 cases keep accumulating in the literature data. Inflammation associated with micro-thrombosis and endothelial dysfunction is the main pathway involved in COVID-19 pathogenesis. Markers of inflammation (CRP, LDH, IL6), endothelial dysfunction (ICAM1, VCAM1, E-selectin), coagulation (fibrinogen) and fibrinolysis (D dimers) are used in assessing disease severity, prognosis and treatment options. Endothelial dysfunction is an important factor in COVID-19 pathogenesis and is associated with disease severity. Endothelial damage and thrombotic complications cause increased endocan levels, and endothelial dysfunction can be assessed using endocan levels, a novel biomarker involved in endothelium events.

In COVID-19 patients, disseminated inflammation and coagulation abnormalities affect endothelial function and induce endocan synthesis and secretion. The aim of this study was the investigation of COVID-19 pathophysiology, focusing on inflammation, thrombosis and endothelial dysfunction. In this context, our study was based on the hypothesis that endocan levels, which are a specific marker of endothelial dysfunction, will be elevated in patients with COVID-19. 

In our study, the endocan levels of patients with COVID-19 were significantly higher (*p* < 0.001) than those of the healthy control group. The results of our study are in accordance with data from previous studies (Gorgun et al.; Pascreau et al.) [17,18]; our results show the median endocan level in the COVID-19 group being 2-fold higher compared to the control group, the same ratio as in the study of Pascreau et al. (2021). Studies concerning the relationship between endocan levels and COVID-19 severity have had contradictory results. Chenevier-Gobeaux et al. (2022) reported a significantly increasing endocan level with COVID-19 severity, while Guzel et al. could not demonstrate statistical differences [19,20]. In our study, the serum levels of endocan did not correlate with disease severity (*p* = 0.117).

Comorbidities, mainly cardiovascular diseases, closely related to inflammation and endothelial injury can influence serum endocan levels. The comorbidity rate among patients in our study group was 82.14%. Patients with cardiovascular diseases had higher levels of serum endocan than patients without, but the difference was not significant. Our data agree with the studies of Gorgun et al. (2022) and Chenevier-Gobeaux et al. (2022); there was no significant difference in serum endocan levels between COVID-19 patients with and without comorbidities [17,21]. 

Cardiovascular pathologies, pulmonary embolism, diabetic vasculopathies, inflammatory rheumatologic diseases and chronic renal failure are associated with elevated serum endocan levels. Tadzic et al. showed that endocan levels tend to decrease with blood pressure reduction after Ca^2+^ blocker administration [22]. In the study group, the main comorbidity was hypertension, and all patients had treatment with different types of medication, including calcium channel blockers. Treatments for lowering blood pressure can influence the serum levels of endocan and endothelial function. In patients with cardiovascular disease, we found a positive correlation between endocan and GGT, while in diabetics, endocan levels correlated with serum ferritin levels.

The serum endocan levels were shown to be related to CT scans: patients with interstitial pneumonia with/without pleural effusion or inflammation (mediastinitis; adenomegaly) had endocan serum levels higher than patients with no signs of pneumonia, but the difference was not significant. High levels of endocan could be related with pulmonary vascular endothelium dysfunction translated in the moderate/severe changes in the radiological findings of CT scans, but more studies are needed. 

Moreover, it was observed that the endocan levels increased in the moderate severity group more than in the mild group, but the levels were lower in the severe group compared to the moderate group; this can be explained by the use of oxygen and therapy with corticoids early in the treatment of the severe cases. In patients with oxygen administration on the first day of admission, the serum level of endocan was lower compared to patients without oxygen therapy. Some patients needed dexamethasone treatment, most of them from the severe disease group. The serum levels of endocan in this group were higher than in patients without dexamethasone treatment and can be related to inflammatory conditions. Endocan is expressed mainly by the vascular endothelial cells, unlike other markers such as fibrinogen or CRP. The literature data are still debatable concerning the precise mechanism of endocan expression in infection, but it is clear that its in vitro expression is stimulated as a result of TNF-α or IL1-β.

The strength of our study is the significant differences in serum endocan levels observed between vaccinated and unvaccinated groups (*p* < 0.001). More than 50% of patients were unvaccinated, and more than 1/3 of them developed a severe form of COVID-19. Vaccination protects against the evolution to severe forms. 

The main findings of our study of endocan levels are as follows: endocan levels were significantly higher in COVID-19 patients than in the control group and can provide additional diagnostic benefits in COVID-19-patient evolution, endocan levels did not correlate with disease severity or the presence of cardiovascular comorbidities and unvaccinated patients had significantly higher levels of endocan than vaccinated patients, which reflects endocan involvement in the complex relationship between immune response, inflammation and endothelial function. Additionally, comparative studies with similar infectious diseases may provide valuable insights into shared pathways and potential therapeutic targets.

The literature includes only a few studies comparing endocan levels in viral and bacterial infections. A single study (Lipinska et al., 2023) compares endocan concentrations in COVID-19 versus bacterial infections, showing higher values for bacterial than for viral sepsis. Vasculopathy associated with COVID-19 appears to be significantly different from that occurring in bacterial infections [23]. Concurrently with the patients in the COVID-19 group, we determined the serum level of endocan in a small number of non-COVID-19 patients. This analysis is the subject of a future study, in which the cohort of non-COVID-19 will have a similar number of patients to the COVID cohort, in order to verify the statistical significance of the data. 

In our study, endocan levels were significantly higher for COVID-19 patients compared to healthy individuals; thus, endocan can be considered as having added diagnostic value for the assessment of COVID-19 patients. The potential value of endocan as a prognostic biomarker might be considered in inflammatory and endothelial-damage circumstances, in correlation with other inflammatory markers. Endocan’s significance as a diagnostic/prognostic factor is under investigation, and further studies, including large/diverse populations, are mandatory to pinpoint its clinical/biochemical use.

The study data correspond and reinforce the previously published results from analyses on groups with a similar number of patients and support the idea of endocan as a diagnostic marker for COVID-19. As far as we know, this is the first study in which endocan has been comparatively determined between vaccinated and unvaccinated patients. The statistical differences in endocan levels between the two groups highlights the protective role of vaccination and the complex relationship between immune response, inflammation and endothelial dysfunction. 

Elevated CRP, LDH, D dimers, fibrinogen and IL6 are significantly associated with severity. In patients with cardiovascular comorbidities, CRP, LDH and fibrinogen were significantly higher and reflect the pro-inflammatory condition of underlying pathologies, which can affect microcirculation and exacerbate the complications of vascular dysfunctions [24]. Thorax CT severity was associated with CRP and D dimer levels and can mark the relationship between inflammation and thrombosis in the pulmonary area. Oxygen-therapy requirement correlated with CRP, LDH and D dimers.

CRP is a non-specific acute-phase inflammation protein, induced by IL6 in the liver, involved in immune host defense by enhancing phagocytosis and activating the complement system [25,26]. CRP is a biomarker of COVID severity and is suitable for guiding therapy. In our study, all patients exhibited a systemic inflammatory response to SARS-CoV-2 infection; in severe cases, the median CRP levels being nearly 40-fold higher than the normal reference upper limit. Our data are in agreement with the study of Smilowitz et al. (2021) on hospitalized patients with elevated CRP concentrations in critical cases [25]. In our study group, serum CRP levels correlated significantly with the severity of disease, presence of cardiovascular diseases, need of oxygen therapy and computer scan severity. Otoshi et al. (2021) demonstrated that age and CRP were independently associated with COVID-19 severity in Japanese patients [27]. Patients with cardiovascular comorbidities have elevated levels of CRP, and they are more likely to display a systemic inflammation. Cremer et al. (2021) detected significantly higher CRP values in patients with cardiovascular diseases in uncomplicated phases of COVID-19 [28]. In our study, for patients with cardiovascular disease, CRP levels positively correlated with AST, ALT, LDH, IL6 and fibrinogen. Similarly to other studies which reported a positive correlation between the severity of thorax CT (quantitative CT scores), our results also revealed a correlation between CRP levels and CT results [29,30].

LDH is involved in the interconversion between pyruvate and lactate in hypoxia (glycolytic pathway upregulated) and multiple organ injury. Increased LDH levels induce metalloproteinase activation and macrophage-mediated angiogenesis [31]. LDH is a marker of cell damage which can indicate an acute or severe tissue injury. There are significant differences in LDH levels between mild and severe cases. In a meta-analysis by Malik et al. (2021), high LDH levels were associated with disease severity [32]. In severe cases from our group, LDH correlated with IL6, WBC and fibrinogen. Serum LDH was higher in subjects with cardiovascular comorbidities, which can indicate vascular damage [33]. Also, patients with hypoxemia had LDH levels significantly higher compared to those with no need of oxygen therapy. Our results are in agreement with studies which indicate that LDH levels over 225 U/L entailed lung involvement and can predict lung injury and severity [25]. Suzuki et al. (2019) suggested that both CRP > 36 mg/L and LDH > 267 U/L are useful for identifying patients with an oxygen requirement [34]. Data from the severe cases in our group confirmed these findings; nearly all patients required oxygen therapy, with median CRP and LDH values being in accordance with the analysis of Suzuki et al.

COVID-19 is a systemic infection with a disturbance of the coagulation system manifested with thrombotic complications and coagulopathy. In our study, coagulation abnormalities were observed, and elevated levels of fibrinogen and D dimers indicated thrombotic complications. Fibrinogen is an acute-phase inflammation protein; elevated levels in COVID-19 patients are associated with damage of the endothelium, hypercoagulability and thrombosis. In our group, fibrinogen was significantly higher in severe than in moderate disease subjects. Similarly, Sui et al. (2021) showed that elevated fibrinogen correlates with disease severity [35]. In COVID-19 patients with cardiovascular disease, fibrinogen and IL6 levels positively correlated. IL6 promotes hemostasis but does not affect fibrinolysis, while the hepatic synthesis of fibrinogen is significantly upregulated by IL6 [36]. In COVID-19 patients with cardiovascular disease, fibrinogen levels were significantly higher than COVID-19 patients without cardiovascular disease. These data parallel the results of Li et al. and reflect a hypercoagulable state and an increased risk of cardiovascular events [37].

D dimers are products of fibrin degradation and can predict venous thromboembolism events in COVID-19 patients and can be used as biomarkers of inflammation and coagulopathy/fibrinolysis. Hypercoagulation and hypoxemia induce high levels of D dimers [38]. In our study, serum levels of D dimers correlated with disease severity and oxygen-therapy requirement. In severe cases, the median concentration was nearly 7-fold higher than the normal upper limit, in agreement with literature reports of a 3- to 4-fold rise in severe cases [39]. Further, D dimers correlated with CT severity; elevated levels were associated with interstitial pneumonia with mediastinitis/pleural effusion. These data mirror the results of Wang et al. (2021) regarding D dimer and CT severity score correlation [40]. In patients with oxygen therapy, D dimers positively correlated with IL6. In unvaccinated patients, D dimers correlated with LDH, reflecting cell damage associated with coagulation abnormalities.

The hyperactivation of the immune system followed by cytokine storm is mediated by IL6, involved in the activation of coagulation cascade and vascular leakage. IL6 signaling contributes to endothelial dysfunction, and serum elevated levels are associated with disease severity [32]. The results of our study indicate an association between the serum level of IL6 and the stage of disease; patients with severe cases have nearly threefold higher serum IL6 levels than those with a mild form. Our data are in agreement with the meta-analysis of Coomes et al. (2020) [8]. Also, in severe cases, IL6 levels correlated with LDH, while in patients with oxygen therapy, it correlated with D dimer levels and fibrinogen.

In the same study group, we also measured the levels of vitamin D, and the determined values correlated positively with endocan in the group with the moderate form and in the patients who did not receive corticosteroid treatment. The results of this analysis will be published in a future article. The literature includes only a few studies on vitamin D–endocan correlation in patients with cardiovascular disease or chronic kidney injury. The results reveal a negative correlation and demonstrate that a deficiency of vitamin D can cause endothelial damage [41,42].

Our findings correspond with previous research suggesting that inflammation, thrombosis and endothelial dysfunction play pivotal roles in COVID-19 pathogenesis, and it is essential to note that the interplay between these pathways is complex. Future research efforts should aim to elucidate the specific molecular mechanisms.

## 4. Materials and Methods

### 4.1. Study Design

The study included 56 patients with a COVID-19 diagnosis, admitted at the National Institute of Infectious Diseases “Prof. Dr. Matei Bals” between 1 September 2022 and 31 January 2023. The control group included 23 healthy subjects from hospital staff. The study was performed according to the principles of the Declaration of Helsinki and was approved by the Medical Ethics Committee of the National Institute of Infectious Diseases “Prof. Dr. Matei Bals” C06884/23.06.2022. COVID-19 diagnosis was confirmed by a polymerase chain reaction test (RT-PCR). The clinical and biological data from patients and controls were gathered, including symptomatology and comorbidities with associated treatment, vaccination status and routine blood biomarkers (CRP, fibrinogen, D dimers, leukocyte and neutrophil count). Imagistic data were collected (computed tomography (CT) and radiologic results); markers of inflammation (IL1, IL6, ferritin) were assessed on blood samples. Patients were classified, (based on the NIH COVID-19 Treatment Guidelines) according to disease severity, from mild (stage 0) to critical (stage 3). Mild cases were characterized by a lack of dyspnea or abnormal imaging (COVID-19 stage 0); moderate cases showed evidence of lower respiratory disease with SpO_2_ > 94% (COVID-19 stage 1); severe cases had SpO_2_ < 94% or respiratory rate >30 or lung infiltrates higher than 50% on CT (COVID-19 stage 2); critical cases presented acute respiratory distress syndrome or septic shock (COVID-19 stage 3).

Blood samples were drawn for the assessment of endocan levels after acquiring informed consent from all study participants. The exclusion criteria were age < 18 years, malignancies, other viral infections (HIV; B, C hepatitis) as well as pregnancy.

### 4.2. Measurement of Serum Endocan Levels

Blood was collected from the patients in the first 48h of admission, in tubes with no anticoagulant. Serum samples were allowed to clot for 2 h at room temperature and were centrifuged at 3000× *g* for 10 min. The serum samples were placed in Eppendorf tubes and stored at −30 °C until they were analyzed. After thawing, the samples were vortexed and Human Endocan/ESM1 levels were analyzed in an ELISA kit in accordance with the instructions of the kit manufacturer (Wuhan Fine Biotech, China, product no. EH0125, batch no. H0125I022). The results were expressed in pg/mL. Calculations were performed using a computer-based curve-fitting software(SPARK10M V4.0.22) and determined through regression analysis (Tecan, Infinite M200 Pro).

Serum glucose (mg/dL), urea (mg/dL), creatinine (mg/dL), bilirubin (mg/dL), creatin kinase MB (CKMB, U/L), creatin kinase (CK, U/L), AST (aspartate aminotransferase U/L), alanine aminotransferase (ALT, U/L), alkaline phosphatase (U/L), lactate dehydrogenase (LDH, U/L), gamma glutamyl transferase (GGT, U/L) and lipase (U/L) levels were measured spectrophotometrically on a VITROS FS3 device. Ferritin (ng/mL) levels were measured using a chemiluminescence method on an ADVIA XPT device, and IL 6 (pg/mL) and TNF-α (pg/mL) were analyzed using the ELISA method.

White blood cell (WBC), lymphocyte and neutrophil counts were estimated on a DXH900 device, and the results were expressed as cells/µL. D dimer levels (ng/mL) were automatically measured on an XACLTOP device.

### 4.3. Statistical Analysis

IBM SPSS Statistics (IBM Corporation, New York, NY, USA) was used for statistical processing. Results were presented as the mean ± standard deviation for normally distributed parameters and as the median [quartile 25; quartile 75] for non-normally distributed. Normality was assessed with a Kolmogorov–Smirnov test. Further, for detecting significant differences among study groups, a one-way ANOVA (with a Games–Howell post hoc test) or T-test was applied and, respectively, a Kruskal–Wallis or Mann–Whitney test, depending on the data distribution and number of groups assessed. To determine significant correlations, the Spearman rank correlation method was used. The level of significance was set at 0.05.

## 5. Conclusions

In this study, patients in different stages of disease severity were included, and their biochemical data were analyzed. Endocan is a novel marker and can be used as a biomarker for the detection of endothelial dysfunction with an involvement in treatment evaluation and disease prognosis.

This study demonstrated that levels of inflammation and coagulation markers correlate with disease severity and the presence of cardiovascular comorbidities. Serum endocan levels were significantly increased in COVID-19 patients compared to healthy controls, but there were no statistically significant differences between the stages of severity. To the best of our knowledge, this is the first study comparing the levels of endocan between unvaccinated and vaccinated groups. Serum endocan levels correlate with vaccination status.

Endocan is a modern biomarker and could emerge as an innovative parameter to evaluate the associated risks in COVID-19 patients; nevertheless, more studies are needed to assess the relationship with inflammatory and thrombosis markers (CRP, fibrinogen, D dimers, IL 6) or CT scans.

The most important limitation of our study was the small number of patients with an impact on the statistical significance on the results. Moreover, the selected patients had some comorbidities (hypertension, diabetes, heart failure, coronary artery disease) with specific treatments that may have affected endocan levels.

## Figures and Tables

**Figure 1 ijms-25-03042-f001:**
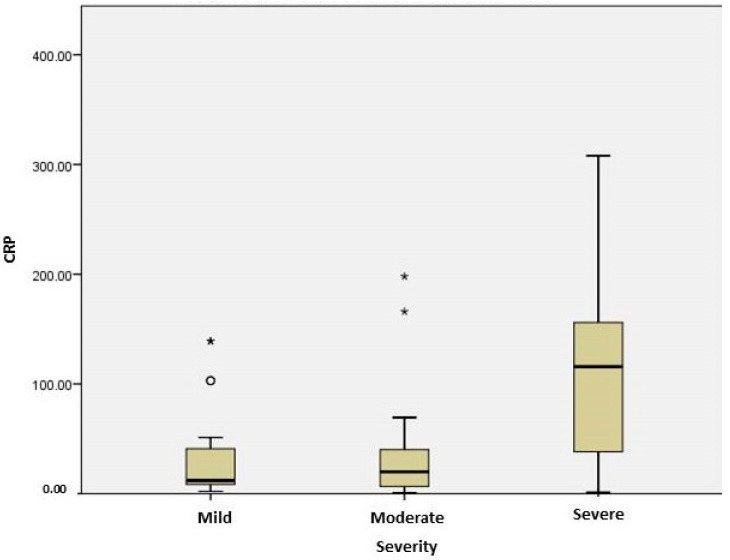
Plasma levels of CRP in patients with different stages of COVID-19. The difference between severe and moderate groups was significant (*p* = 0.008). The box plots represent the median values (midpoint) with an interquartile range between the 25th and 75th percentiles (box), along the minimum and maximum values and outliers and extreme values (*).

**Figure 2 ijms-25-03042-f002:**
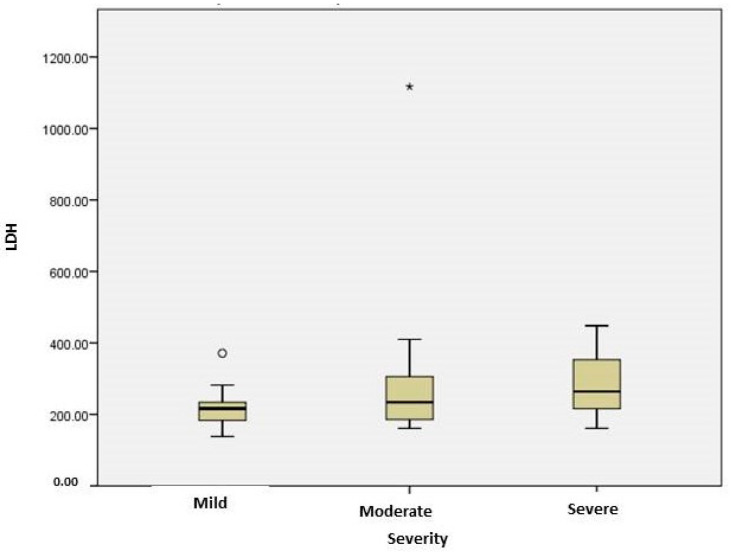
Plasma levels of LDH in patients with different stages of COVID-19. The difference between severe and mild groups was significant (*p* = 0.035). The box plots represent the median values (midpoint) with an interquartile range between the 25th and 75th percentiles (box), along the minimum and maximum values and outliers and extreme values(*).

**Figure 3 ijms-25-03042-f003:**
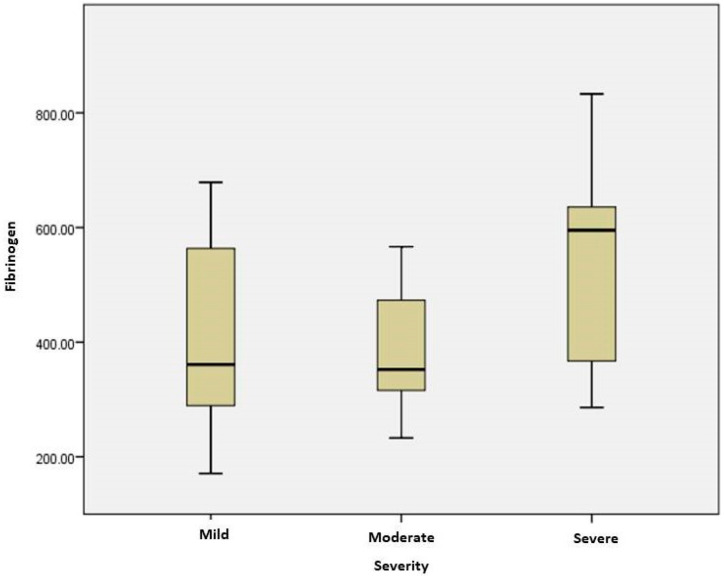
Plasma levels of fibrinogen in patients with different stages of COVID19. The difference between severe and moderate groups was significant (*p* = 0.021). The box plots represent the median values (midpoint) with an interquartile range between the 25th and 75th percentiles (box), along the minimum and maximum values.

**Figure 4 ijms-25-03042-f004:**
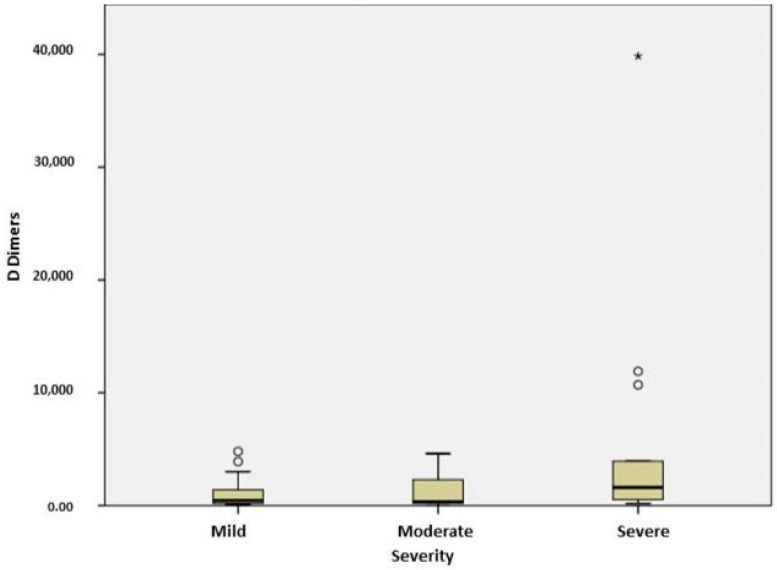
Plasma levels of D dimers in patients with different stages of COVID-19. The difference between severe and mild groups was significant (*p* = 0.046). The box plots represent the median values (midpoint) with an interquartile range between the 25th and 75th percentiles (box), along the minimum and maximum values and outliers and extreme values(*).

**Table 1 ijms-25-03042-t001:** Characteristics of the study group.

Demographics	
Age (years)	71.48 ± 14.75
Male n (%)	27 (48.21%)
Female n (%)	29 (51.79%)
Comorbidities	
Cardiovascular disease, n (%)	46 (82.14%)
Hypertension	36 (64.28%)
Heart failure	10 (17.85%)
Arrhythmias	14 (25%)
Overweight/Obesity	14 (25%)
Diabetes mellitus	10 (17.85%)
Hyperlipidemia	10 (17.85%)
Chronic renal failure	7 (12.5%)
Chronic obstructive pulmonary disease	4 (7.14%)
Thorax CT finding	
Interstitial pneumonia with pleural effusion/mediastinitis	18 (32.14%)
Interstitial pneumonia	25 (44.64%)
No finding	9 (16.07%)
Treatment at admission	
Oxygen therapy	13 (23.31%)
Dexamethasone	15 (26.78%)
Vaccinating status	
Vaccinated	25 (44.64%)
Unvaccinated	31 (55.36%)
Clinical outcome	
Discharged with recovery	54 (96.42%)
Death	2 (3.58%)

**Table 2 ijms-25-03042-t002:** Laboratory data of the study group, expressed as the mean and standard deviation for values with a normal distribution and the median (quartile 25, quartile 75) for those without a normal distribution.

Parameters	Value
Glucose (mg/dL)	106.5 [98; 126.25]
Urea (mg/dL)	36.5 [27; 54.5]
Creatinine (mg/dL)	0.9 [0.7; 1.18]
Bilirubin (mg/dL)	0.6 [0.5; 0.8]
CKMB (U/L)	12 [8; 19]
CK (U/L)	71 [42; 116]
ALT (U/L)	25 [19.25; 39.5]
AST (U/L)	32 [26.25; 46]
LDH (U/L)	234 [187; 305]
Alkaline phosphatase (U/L)	78 [56; 94]
GGT (U/L)	33 [25; 51]
Lipase (U/L)	117.5 [55; 193.75]
CRP (mg/L)	27.6 [7.52; 85.3]
Ferritin (ng/mL)	303.25 [187.48; 539.2]
IL6 (pg/mL)	71.9 [24.5; 206.1]
TNF-α (pg/mL)	67.94 ± 37.17
IL1 (pg/mL)	13.9 [1.49; 40.3]
PAI1(pg/mL)	267.66 ± 153.62
WBC × 1000/µL	7.4 [4.56; 9.28]
Neutrophil × 1000/µL	5.5 ± 2.59
Lymphocyte × 1000/µL	0.85 [0.69; 1.28]
Hemoglobin (g/dL)	12.53 ± 1.35
Platelets × 1000/µL	193 [155; 260]
D dimers (ng/mL)	530 [214.5; 3100]
Fibrinogen (mg/dL)	367.17 [307.5; 575.7]
Endocan (pg/mL)	77.21 ± 31.4

**Table 3 ijms-25-03042-t003:** The demographic, clinical characteristics and laboratory data of patients, depending on the severity of the disease.

Demographics	Mild Cases	Moderate Cases	Severe Cases	*p* Value
Age (years)	63.5 [51.25; 78]	78.5 [78; 83]	83 [76.75; 85.5]	
Males	2 (13.33%)	16 (64%)	9 (56.25%)	
Females	13 (86.67%)	9 (36%)	7 (43.75%)	
Comorbidities				
Cardiovascular disease	5 (33.33%)	15 (60%)	16 (100%)	
Hypertension	4 (26.67%)	13 (52%)	13 (81.25%)	
Overweight/Obesity	3 (20%)	6 (24%)	5 (31.25%)	
Diabetes mellitus	1 (6.66%)	4 (16%)	4 (25%)	
Vaccinating status				
Vaccinated	5 (33.3%)	14 (56%)	6 (37.75%)	
Unvaccinated	10 (66.67%)	11 (44%)	9 (62.25%)	
Treatment on admission				
Oxygen therapy	0	1 (4%)	13 (81.25%)	
Dexamethasone used	3 (20%)	2 (8%)	9 (56.25%)	
Biochemical parameter				
Urea(mg/dL)	26.5 [22.25; 37]	36 [28.75; 50]	56.5 [36.5; 86.5]	<0.001
Creatinine (mg/dL)	0.75 [0.7; 1]	1 [0.78; 1.2]	1.1 [0.78; 1.83]	0.041
LDH(U/L)	216.5 [177.25; 236]	234 [178; 307]	264 [213.5; 353.5]	0.04
C reactive protein (mg/L)	11.95 [7.57; 46.08]	19.9 [6.4; 41.58]	115.8 [36.78; 156.7]	0.004
IL6 (pg/mL)	108.05 [25.9; 218.4]	42.1 [11.4; 93.1]	343.7 [94.43; 645.9]	0.012
Alkaline phosphatase (U/L)	78 [61.5; 110]	70 [55; 94.75]	84 [55.75; 94.25]	0.804
GGT (U/L)	32 [21.5; 56]	33 [27.5; 73.25]	37.5 [24.5; 51]	0.813
Red blood cells	4.71 [4.5; 4.95]	4.35 [3.85; 4.49]	4.06 [3.7; 4.76]	0.004
Hemoglobin (g/dL)	13.2 [11.63; 13.78]	12.6 [11.68; 13.43]	11.9 [11.2; 14]	0.656
Prothrombin time (s)	12.05 [11.2; 13.13]	12.1 [11.38; 14.13]	14 [12.45; 15.58]	0.044
D dimers (ng/mL)	448 [190; 1427.5]	372 [190; 2525]	1600 [464.5; 7322]	0.046
Fibrinogen (mg/dL)	361 [283.65; 596.26]	352.25 [307.8; 480.91]	595.37 [356.27; 637.22]	0.021
Endocan (pg/mL)	74.28 ± 35.97	87.2 ± 29.2	65.7 ± 24.35	0.117

**Table 4 ijms-25-03042-t004:** Differences in inflammatory markers as well as endocan levels between patients with and without cardiovascular disease.

	With Cardiovascular Disease (46)	Without Cardiovascular Disease (10)	*p* Value
Alkaline phosphatase (U/L)	84 [59; 112.5]	67 [54.75; 80.75]	0.034
Lactate dehydrogenase (U/L)	251 [202.75; 320]	211 [160.5; 240]	0.030
C reactive protein (mg/L)	38.2 [11.35; 121]	10.2 [6.28; 26.5]	0.006
Fibrinogen (mg/dL)	472.98 [332.26; 603.51]	289.29 [257.22; 375.85]	0.002
Endocan (pg/mL)	79 ± 30.39	73.72 ± 34.02	0.557

**Table 5 ijms-25-03042-t005:** Serum levels of endocan for different comorbidities, therapy used at admission and vaccination status.

	Endocan Levels (pg/mL)		*p* Value
	Absent	Present	
Comorbidity	77.08 ± 23.99	77.24 ± 33.08	0.988
Diabetes	77.2 ± 31.99	77.28 ± 30.31	0.994
Cardiovascular disease	73.72 ± 34.02	79 ± 30.39	0.557
Obesity	79.12 ± 31.27	70.89 ± 32.51	0.414
Oxygen need	78.75 ± 33.02	71.55 ± 25.32	0.424
Dexamethasone treatment	75.47 ± 32.22	81.96 ± 29.81	0.499
Vaccination	89.63 ± 28.5	61.81±28.41	0.001

## Data Availability

Data are contained within the article.

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
