# Peer review of "Investigation of Serum Endocan Levels in SARS-CoV-2 Patients"

_ijms, 2024, doi:10.3390/ijms25053042_

Round 1

Reviewer 1 Report

Comments and Suggestions for Authors

Viral infection induces a dysregulated hyperactivation of the immune system, a prothrombotic state manifested as micro-thrombosis, disseminated inflammation and vascular endothelium injuries, mainly in pulmonary area.

Please provide references.

SARS COV 2 infection causes endothelial dysfunction by direct viral effect, through cytokine release, oxidative stress, coagulation disturbance and inflammation with leucocyte adhesion, dysregulated immune cells response.

Please provide references.

Viral infection induces  endothelial dysfunction through activating coagulation pathways, followed by a coagulation imbalance. Elevated D dimers, increased fibrinogen and enhanced platelet activation indicate a dysfunctional endothelial response to viral infection.

Please cite the following article in this context.

Deep phenotyping of the lipidomic response in COVID-19 and non-COVID-19 sepsis, Meng et.al., Clinical and Translational Medicine, 2023.

Serum levels of interleukin 6 (IL6) are elevated in inflammatory conditions and are associated with  COVID 19 severity. Activation of IL6 receptors induce upregulation of adhesion molecules with enhanced leucocyte adherence and extravasation in vascular wall.

Please provide references.

Patients with  hypoxemia had LDH levels significantly higher compared to those without oxygen  therapy need (p=0,000)

Correct the p value.

What is the status of sPLA2 in the covid patients? Does it correlate with endocan?

Since there was no correlation with disease severity, does it mean it is specific to the presence of the infection? The authors should compare their results with other viral and bacterial infections to explore if that is COVID-specific.

How would the levels of endocan help in the diagnosis and prognosis of covid-19? The authors should comment on that.

Please make the discussion in line with the novel finding of the study.

iThenticate report states 30% match. The authors should some part of the text.

Author Response

We thank the reviewers for their efforts and their insightful comments, which helped us a lot in improving the manuscript. Please see below our point-to-point responses

Reviewer 1

  1. Viral infection induces a dysregulated hyperactivation of the immune system, a prothrombotic state manifested as micro-thrombosis, disseminated inflammation and vascular endothelium injuries, mainly in pulmonary area.

Please provide references.

Answer:

The manuscript was updated with new reference:

  1. Yuki, M. Fujiogi, and S. Koutsogiannaki, “COVID-19 pathophysiology: A review,” Clinical Immunology, vol. 215. Academic Press Inc., Jun. 01, 2020. doi: 10.1016/j.clim.2020.108427.

  1. SARS COV 2 infection causes endothelial dysfunction by direct viral effect, through cytokine release, oxidative stress, coagulation disturbance and inflammation with leucocyte adhesion, dysregulated immune cells response.

Please provide references.

Answer

The manuscript was updated with new reference:

  1. wen Xu, I. Ilyas, and J. ping Weng, “Endothelial dysfunction in COVID-19: an overview of evidence, biomarkers, mechanisms and potential therapies,” Acta Pharmacologica Sinica, vol. 44, no. 4. Springer Nature, pp. 695–709, Apr. 01, 2023. doi: 10.1038/s41401-022-00998-0.
  2. Viral infection induces endothelial dysfunction through activating coagulation pathways, followed by a coagulation imbalance. Elevated D dimers, increased fibrinogen and enhanced platelet activation indicate a dysfunctional endothelial response to viral infection.

Please cite the following article in this context.

Deep phenotyping of the lipidomic response in COVID-19 and non-COVID-19 sepsis, Meng et.al., Clinical and Translational Medicine, 2023.

Answer

The manuscript was updated with the suggested reference.

  1. Serum levels of interleukin 6 (IL6) are elevated in inflammatory conditions and are associated with COVID 19 severity.

Answer

The manuscript was updated with new reference:

  1. A. Coomes and H. Haghbayan, “Interleukin-6 in Covid-19: A systematic review and meta-analysis,” Reviews in Medical Virology, vol. 30, no. 6. John Wiley and Sons Ltd, pp. 1–9, Nov. 01, 2020. doi: 10.1002/rmv.2141.

Activation of IL6 receptors induce upregulation of adhesion molecules with enhanced leucocyte adherence and extravasation in vascular wall.

Please provide references.

Answer

The manuscript was updated with new reference:

  1. Jin, W. Ji, H. Yang, S. Chen, W. Zhang, and G. Duan, “Endothelial activation and dysfunction in COVID-19: from basic mechanisms to potential therapeutic approaches,” Signal Transduction and Targeted Therapy, vol. 5, no. 1. Springer Nature, Dec. 01, 2020. doi: 10.1038/s41392-020-00454-7.

  1. Patients with hypoxemia had LDH levels significantly higher compared to those without oxygen therapy need (p=0,000)

Correct the p value.

Answer

Thank you for the comment. The manuscript was updated according to your recommendation.

  1. What is the status of sPLA2 in the covid patients? Does it correlate with endocan?

Answer:

Thank you for valuable comment. In this study, we did not record dosages for sPLA2, but the analysis of this parameter could be the subject of a future study. Lipase values were determined, but were no correlations with endocan levels. According to Farooqui et al. (2023) review, sPLA2 level correlates with severity in COVID 19 and activity of type sPLA2 IIA is linked to cytokine storm and coagulopathy. There are no studies about sPLA2 – endocan relation, but is a very interesting subject for future research, particularly both molecules activate in endothelial cells.

  1. Since there was no correlation with disease severity, does it mean it is specific to the presence of the infection? The authors should compare their results with other viral and bacterial infections to explore if that is COVID-specific.

Answer:

Thank you for the valuable suggestion. The following parts was added to the discussion section

Literature includes only a few studies comparing endocan levels in viral and bacterial infections. A single study (Lipinska et al.,2023) compares endocan concentrations in COVID 19 versus bacterial infections, showing higher values for bacterial than for viral sepsis. The vasculopathy associated with COVID 19 appears to be significantly different from that occurring in bacterial infections. Concurrently with the patients in the COVID 19 group, we determined the serum level of endocan in a small number of non-COVID 19 patients. This analysis is the subject of a future study, in which the cohort of non-COVID 19 will have a similar number of patients to the COVID cohort, in order to verify the statistical significance of the data.

  1. How would the levels of endocan help in the diagnosis and prognosis of covid-19? The authors should comment on that.

Answer:

Thank you for the comment. The following parts was added to the discussion section

Endocan levels were significantly higher in COVID 19 patients compared to healthy individuals and can have added diagnostic benefits in assessing COVID 19 patients. The potential ability of endocan as a prognostic biomarker is in inflammatory conditions with endothelial damage, related with other inflammatory markers. Endocan may be associated with COVID 19 severity in cardiovascular disease. Endocan significance as a diagnostic or prognostic factor in inflammatory with endothelial damage diseases is under investigation, and further studies with a larger study population are warranted to determine its clinical use.

  1. Please make the discussion in line with the novel finding of the study.

Answer

Thank you for your suggestion. The following parts was added to the discussion section

Prior research has identified endocan as an indicator of endothelial dysfunction and inflammation in various diseases, but its relevance in the context of COVID19 should be investigate. Analysis of the correlation between endocan levels and diagnosis, disease severity, prognosis, biochemical parameters and vaccination status contributes to the evidence regarding endocan’s role in inflammation and endothelial dysfunction in COVID 19. The study data corresponds and reinforces previously published results from analyses on groups with similar number of patients, and support the idea of endocan as a diagnostic marker in COVID 19. As far as we know, this is the first study in which endocan has been comparatively determined between vaccinated and unvaccinated patients. The statistical differences in endocan levels between the two groups highlights the protective role of vaccination, the complex relationship between immune response, inflammation and endothelial dysfunction.

Please revise your manuscript, which some paragraphs are similar to the
self-published papers in the main text (highlight in attachment) during
revision.

Thank you for revision and suggestion, minor changes were made according to your recommendation regarding similarity, these paragraphs indicate a standard, neutral description of material and methods and statistical analysis.

Reviewer 2 Report

Comments and Suggestions for Authors

The topic is interesting and the paper is quite well written. Nevertheless, in my opinion, some parts need to be improved, I have some comments:

1- Conclusion: Endocan can be considered a novel biomarker for the detection of inflammation and endothelial dysfunction risk in COVID 19 patients. Abstract might be beneficial to include a sentence that briefly summarizes the key findings of the study. This can provide readers with a quick overview of the research. 

2- 1. Introduction 29 Coronavirus disease 2019 (COVID 19) is a clinical syndrome caused by infection 30 with a strain of coronavirus (SARS COV 2), called severe acute respiratory syndrome 31 (SARS)-associated coronavirus and is linked, for an important percentage of the hospi- 32 talized patients, to long-term consequences and mortality [1]. Viral infection induces a 33 dysregulated hyperactivation of the immune system, a prothrombotic state manifested 34 as micro-thrombosis, disseminated inflammation and vascular endothelium injuries, 35 mainly in pulmonary area. Although COVID 19 was initially considered a viral pneu- 36 monia leading to acute respiratory failure, the clinical, laboratory, and postmortem 37 findings suggest that altered endothelial function is a contributing factor of its patho- 38 physiology [2]. SARS COV 2 infection causes endothelial dysfunction by direct viral ef- 39 fect, through cytokine release, oxidative stress, coagulation disturbance and inflamma- 40 tion with leucocyte adhesion, dysregulated immune cells response. Although the Authors described in detail the findings from the included references, there are several relevant works/reviews, including most recently published which should be added and discussed by the Authors to ameliorate the paragraph:

a- Quantitative Computed Tomography Lung COVID Scores with Laboratory Markers: Utilization to Predict Rapid Progression and Monitor Longitudinal Changes in Patients with Coronavirus 2019 (COVID-19) Pneumonia. Biomedicines. 2024;12(1):120.  doi:10.3390/biomedicines12010120

b- Radiological-pathological signatures of patients with COVID-19-related pneumomediastinum: is there a role for the Sonic hedgehog and Wnt5a pathways?. ERJ Open Res. 2021;7(3):00346-2021.  doi:10.1183/23120541.00346-2021

c- Ventilatory associated barotrauma in COVID-19 patients: A multicenter observational case control study (COVI-MIX-study). Pulmonology. 2023;29(6):457-468. doi:10.1016/j.pulmoe.2022.11.002

3- The aim of our study was to evaluate the relationship of endocan levels in COVID 102 19 hospitalized patients with disease diagnosis and prognosis, as well as with the bio- 103 chemical profile of patients and their vaccination status in order to establish the rele- 104 vance of endocan assessment as a prognostic marker for SARS COV 2 patients. Please improve the description of this part and underline the novelty of the study.

4- 2. Results 106 2.1. Clinical Characteristics of COVID19 patients. Please, underline the most important statistically significant results to clarify the data. 

5- 3. Discussion 227 Numerous studies addressing COVID 19 pathophysiology were published, but 228 disease mechanism is still not completely elucidated, especially since reports regarding 229 COVID 19 cases keep accumulating in literature data. Inflammation associated with mi- 230 cro-thrombosis and endothelial dysfunction are the main pathways involved in COVID 231 19 pathogenesis.... The discussion section needs to be improved.  It could be interesting to record the aim of the study. It is necessary to be more concise in the presentation of the facts, clarifying the results obtained and comparing them with previous or similar studies. However, it is interesting to answer the questions that arise from these results, backed up by published literature. 

Comments on the Quality of English Language

 Minor changes of English language are required

Author Response

We thank the reviewers for their efforts and their insightful comments, which helped us a lot in improving the manuscript. Please see below our point-to-point responses

Reviewer 2

The topic is interesting and the paper is quite well written. Nevertheless, in my opinion, some parts need to be improved, I have some comments:

  1. Conclusion: Endocan can be considered a novel biomarker for the detection of inflammation and endothelial dysfunction risk in COVID 19 patients. Abstract might be beneficial to include a sentence that briefly summarizes the key findings of the study. This can provide readers with a quick overview of the research. 

Answer:

Thank you for the valuable comment. The manuscript was updated according to your recommendation.

Results revealed significantly elevated serum endocan levels in COVID-19 patients compared to the control group, with a correlation observed between endocan concentration and vaccination status. These findings suggest that endocan may serve as a novel biomarker for detecting inflammation and endothelial dysfunction risk in COVID-19 patients. There was no significant relationship between serum endocan levels and disease severity and presence of cardiovascular diseases

  1. Introduction 29 Coronavirus disease 2019 (COVID 19) is a clinical syndrome caused by infection 30 with a strain of coronavirus (SARS COV 2), called severe acute respiratory syndrome 31 (SARS)-associated coronavirus and is linked, for an important percentage of the hospi- 32 talized patients, to long-term consequences and mortality [1]. Viral infection induces a 33 dysregulated hyperactivation of the immune system, a prothrombotic state manifested 34 as micro-thrombosis, disseminated inflammation and vascular endothelium injuries, 35 mainly in pulmonary area. Although COVID 19 was initially considered a viral pneu- 36 monia leading to acute respiratory failure, the clinical, laboratory, and postmortem 37 findings suggest that altered endothelial function is a contributing factor of its patho- 38 physiology [2]. SARS COV 2 infection causes endothelial dysfunction by direct viral ef- 39 fect, through cytokine release, oxidative stress, coagulation disturbance and inflamma- 40 tion with leucocyte adhesion, dysregulated immune cells response. Although the Authors described in detail the findings from the included references, there are several relevant works/reviews, including most recently published which should be added and discussed by the Authors to ameliorate the paragraph:

a- Quantitative Computed Tomography Lung COVID Scores with Laboratory Markers: Utilization to Predict Rapid Progression and Monitor Longitudinal Changes in Patients with Coronavirus 2019 (COVID-19) Pneumonia. Biomedicines. 2024;12(1):120.  doi:10.3390/biomedicines12010120

b- Radiological-pathological signatures of patients with COVID-19-related pneumomediastinum: is there a role for the Sonic hedgehog and Wnt5a pathways?. ERJ Open Res. 2021;7(3):00346-2021.  doi:10.1183/23120541.00346-2021

c- Ventilatory associated barotrauma in COVID-19 patients: A multicenter observational case control study (COVI-MIX-study). Pulmonology. 2023;29(6):457-468. doi:10.1016/j.pulmoe.2022.11.002

Answer

Thank you for you suggestion. Similarly to other studies, which reported a positive correlation between severity thorax CT (quantitative CT scores), our results also revealed a correlation between CRP levels and CT results . The manuscript was updated with new references.

Quantitative Computed Tomography Lung COVID Scores with Laboratory Markers: Utilization to Predict Rapid Progression and Monitor Longitudinal Changes in Patients with Coronavirus 2019 (COVID-19) Pneumonia. Biomedicines. 2024;12(1):120.  doi:10.3390/biomedicines12010120

- Ventilatory associated barotrauma in COVID-19 patients: A multicenter observational case control study (COVI-MIX-study). Pulmonology. 2023;29(6):457-468. doi:10.1016/j.pulmoe.2022.11.002

Radiological-pathological signatures of patients with COVID-19-related pneumomediastinum: is there a role for the Sonic hedgehog and Wnt5a pathways?. ERJ Open Res. 2021;7(3):00346-2021.  doi:10.1183/23120541.00346-2021

  1. The aim of our study was to evaluate the relationship of endocan levels in COVID 102 19 hospitalized patients with disease diagnosis and prognosis, as well as with the bio- 103 chemical profile of patients and their vaccination status in order to establish the rele- 104 vance of endocan assessment as a prognostic marker for SARS COV 2 patients. Please improve the description of this part and underline the novelty of the study.

Thank you for the valuable comment. The manuscript was updated according to your recommendation

Previous studies endocan as a marker of endothelial dysfunction and inflammation in cardiovascular diseases, diabetic vasculopathies, sepsis lung and renal diseases, malignicies. COVID 19 is an endothelial dependent disease, systemic inflammation and microangiopathy induce endothelial activation and dysfunction. In this context, endocan should be investigated as a potential marker of endothelial damage and its role in inflammatory and thrombotic events should be elucidated. Our analysis contributes to the evidence regarding endocan’s role in inflammation and endothelial dysfunction in COVID 19. The study data are in agreement and reinforces previously published results highlighting the idea of endocan as a diagnostic marker in COVID 19. As far as we know, this is the first study in which endocan has been comparatively determined between vaccinated and unvaccinated patients. The significantly differences of endocan serum levels between the two groups indicate the protective role of vaccination, the complex relationship between immune response, inflammation and endothelial function.

  1. Results 106 2.1. Clinical Characteristics of COVID19 patients. Please, underline the most important statistically significant results to clarify the data. 

Answer:

Thank you for the suggestion. The manuscript was updated according to your recommendation

Thank you for important suggestion. The majority of patients have an increased level of CRP and it correlates with severity, presence of cardiovascular comorbidities, thorax CT severity and oxygen therapy requirement. Elevated LDH, D dimers, fibrinogen and IL6 are significantly associated with severity. In patients with cardiovascular comorbidities, CRP, LDH and fibrinogen were significantly higher and reflect the pro inflammatory condition of underlying pathologies, which can affect microcirculation and exacerbate complications of vascular dysfunctions. Thorax CT severity was associated with CRP and D dimers levels and can mark the relationship between inflammation and thrombosis in pulmonary area. Oxygen therapy requirement correlates CRP, LDH and D dimers.

  1. Discussion 227 Numerous studies addressing COVID 19 pathophysiology were published, but 228 disease mechanism is still not completely elucidated, especially since reports regarding 229 COVID 19 cases keep accumulating in literature data. Inflammation associated with mi- 230 cro-thrombosis and endothelial dysfunction are the main pathways involved in COVID 231 19 pathogenesis.... The discussion section needs to be improved.  It could be interesting to record the aim of the study. It is necessary to be more concise in the presentation of the facts, clarifying the results obtained and comparing them

Answer:

Thank you for the suggestion. The manuscript was updated according to your recommendation.

The aim of this study was investigation of the COVID 19 pathophysiology, focusing on the inflammation, micro-thrombosis and endothelial dysfunction. In this context our study based on the hypothesis that endocan levels, which is a specific marker of endothelial dysfunction will be elevated in patients with COVID 19. The main findings of our study of endocan levels are: endocan levels are significantly higher in COVID 19 patients than in control group and can provide additional diagnostic benefits in COVID 19 patients evalution; endocan levels doesn’t correlate with disease severity or presence of cardiovascular comorbidities; unvaccinated patients have significantly higher levels of endocan than vaccinated patients and reflects endocan involvment in complex relationship between viral infection, immune response and endothelial function. Our findings correspond with previous research suggesting that inflammation, micro-thrombosis, and endothelial dysfunction play pivotal roles in COVID-19 pathogenesis and it is essential to note that the interplay between these pathways is complex. Future research efforts should aim to elucidate the specific molecular mechanisms. Additionally, comparative studies with similar infectious diseases may provide valuable insights into shared pathways and potential therapeutic targets.

Please revise your manuscript, which some paragraphs are similar to the
self-published papers in the main text (highlight in attachment) during
revision.

Thank you for revision and suggestion, minor changes were made according to your recommendation regarding similarity, these paragraphs indicate a standard, neutral description of material and methods and statistical analysis.

Reviewer 3 Report

Comments and Suggestions for Authors

Content suggestions:

1.         The Authors did not mention the effect of vitamins D, C, zinc or other supportive care on endocan levels. Is there any correlation ?

2.         Can the Authors add the details about thromboprophylaxis and its association with D-dimers, fibrinogen and endocan levels ? I suppose that they used LMWH...

I sincerely appreciate each new information about COVID-19, as this is life-threatening infection. Thus, after the implementation of the responses of the Authors to the questions of the reviewers, the article might be accepted for publication.

Author Response

We thank the reviewers for their efforts and their insightful comments, which helped us a lot in improving the manuscript. Please see below our point-to-point responses

Reviewer3

  1. The Authors did not mention the effect of vitamins D, C, zinc or other supportive care on endocan levels. Is there any correlation ?

Answer:

Thank you for the valuable comment.

In the same study group, we also measured the levels of vitamin D and the determined values correlates positively with endocan in the group with moderate form and in the patients who did not receive corticosteroid treatment. The results of this analysis will be published in a future article. Literature includes only a few studies on vitamin D-endocan correlation in patients with cardiovascular disease or chronic kidney injury. The results reveal a negatively correlation and demonstrate that deficiency of vitamin D can cause endothelial damage. . The manuscript was updated with new references:

-Severe vitamin D deficiency is associated with endothelial inflammation in healthy individuals even in the absence of subclinical atherosclerosis

M Kose 1N SenkalT TukekT CebeciS C AtalarM AltinkaynakH AriciS GencY CatmaM KocaagaA MedetalibeyogluS Emet; European Review for Medical and Pharmacological Sciences 2022; 26: 7046-7052

Vitamin D Treatment Effect on Serum Endocan and High-Sensitivity C-Reactive Protein Levels in Renal Transplant Patients

Omer Atis 1Mustafa Keles 2Erdem Cankaya 3Hasan Dogan 1Hulya Aksoy 4Fatih Akcay

Progress in Transplantation  2016, ,DOI: 10.1177/1526924816664086;

  1. Can the Authors add the details about thromboprophylaxis and its association with D-dimers, fibrinogen and endocan levels ? I suppose that they used LMWH...

        Answer

Thank you for interesting suggestion. The manuscript was updated according to your recommendation

Thromboprophylaxis was used in moderate and severe cases with LMWH or new oral anticoagulation (apixaban).

I sincerely appreciate each new information about COVID-19, as this is life-threatening infection. Thus, after the implementation of the responses of the Authors to the questions of the reviewers, the article might be accepted for publication.

Please revise your manuscript, which some paragraphs are similar to the
self-published papers in the main text (highlight in attachment) during
revision.

Thank you for revision and suggestion, minor changes were made according to your recommendation regarding similarity, these paragraphs indicate a standard, neutral description of material and methods and statistical analysis.

Round 2

Reviewer 2 Report

Comments and Suggestions for Authors

The manuscript has been improved. I have no further comments.

Comments on the Quality of English Language

Minor changes of English language are required